# During Water Stress, Fertility Modulated by ROS Scavengers Abundant in Arabidopsis Pistils

**DOI:** 10.3390/plants12112182

**Published:** 2023-05-31

**Authors:** Ya-Ying Wang, Donald J. Head, Bernard A. Hauser

**Affiliations:** 1Department of Biology, University of Florida, Gainesville, FL 32611, USA; yaying.wang@gmail.com (Y.-Y.W.); djhead@me.com (D.J.H.); 2Plant Molecular and Cellular Biology Program, University of Florida, Gainesville, FL 32611, USA

**Keywords:** ovules, plant stress, fertility, seed formation, reactive oxygen

## Abstract

Hours after watering plants with 75 mM NaCl, the water potential of reproductive structures precipitously decreases. In flowers with mature gametes, this change in water potential did not alter the rate of fertilization but caused 37% of the fertilized ovules to abort. We hypothesize that the accumulation of reactive oxygen species (ROS) in ovules is an early physiological manifestation associated with seed failure. In this study, we characterize ROS scavengers that were differentially expressed in stressed ovules to determine whether any of these genes regulate ROS accumulation and/or associate with seed failure. Mutants in an iron-dependent superoxide dismutase (FSD2), ascorbate peroxidase (APX4), and three peroxidases (PER17, PER28, and PER29) were evaluated for changes in fertility. Fertility was unchanged in *apx4* mutants, but the other mutants grown under normal conditions averaged a 140% increase in seed failure. In pistils, *PER17* expression increases three-fold after stress, while the other genes decreased two-fold or more following stress; this change in expression accounts for differences in fertility between healthy and stressed conditions for different genotypes. In pistils, H_2_O_2_ levels rose in *per* mutants, but only in the triple mutant was there a significant increase, indicating that other ROS or their scavengers be involved in seed failure.

## 1. Introduction

Reactive oxygen species (ROS), such as superoxide (O_2_^−^), hydroxyl radicals (•OH), and hydrogen peroxide (H_2_O_2_), are a group of highly active small molecules that can be naturally produced during cellular metabolism. In plants, ROS are mainly produced in chloroplasts, mitochondria, and peroxisomes during photosynthesis, respiration, or other metabolic processes [1,2]. Low to moderate amounts of ROS signal different responses and regulate many biological processes, e.g., development and defense responses [3,4]. Excessive ROS, however, change the redox homeostasis and leads to oxidative stress [5,6,7]. A classic example of redox signaling is the activation of lumen enzymes after illumination. Recently, mutants affecting thioredoxin and glutathione reduction changed auxin transport, which interfered with floral initiation, showed vasculature defects, and altered root morphology [8]. Furthermore, the specific isoforms that interact to form an active auxin receptor show differential responses to osmotic and salt stress and can lead to different ROS levels [9]. Among various ROS in plants, H_2_O_2_ is generally considered to be the main molecule that serves as a long-range signaling molecule because it is relatively stable and can diffuse relatively rapidly crossing membranes. ROS can damage proteins, break down fatty acids, damage DNA, and induce programmed cell death (PCD) [6,10,11]. Understanding how plants regulate ROS levels can permit hypotheses of mechanisms to prevent excessive ROS production.

In plant cells, there are several ROS-scavenging enzymes to maintain redox homeostasis. Superoxide dismutases (SOD) convert superoxide to H_2_O_2_ and oxygen [12]. Catalases (CAT) break down H_2_O_2_ into water and oxygen [13]. Glutathione peroxidases (GPX) reduce H_2_O_2_ by oxidizing glutathione (GSH), and glutathione reductases (GR) regenerate GSH utilizing NAD(P)H [14]. Ascorbate peroxidases (APX), which are members of class I peroxidases, reduce H_2_O_2_ by oxidizing ascorbate [15]. Each of these ROS scavengers is found in different cell compartments and locations [1,5,16]. The class III heme-containing peroxidases (PER or PRX) are unique to plants and make up a large gene family. In Arabidopsis, there are 73 of these annotated peroxidases [17,18]. Many of these peroxidases are predicted to localize to the cell wall or vacuole and affect lignification, defense responses, hormone metabolism, salt tolerance, and development [19,20]. Horseradish peroxidase (HRP) is the best-known class III peroxidase. The radical products generated after this catalytic process trigger pathogen defense responses [21]. The functions of most of these class III peroxidases, however, remain unknown.

ROS are recognized as key modulators of PCD [22,23]. Many environmental stresses, such as heat shock, chilling, salt, high light, and pathogen attack, cause ROS accumulation [7,13,24,25,26,27]. ROS accumulation was observed in salt-stressed ovules and was hypothesized to cause high abortion rates in Arabidopsis [26,28]; fertility is reduced because seeds arise from ovules.

Research into stress effects on plant reproduction usually attributes decreases in seed set to pollen defects or pollen tube growth, but few studies evaluated the effects of stress on ovules or embryos. In natural populations, pollen limitation reduces fertility for 60% of plant species. Pollen becomes limiting when plant stress induces pollen abortion or the development of low-quality pollen where the pollen tube fails to transmit the sperm nuclei to the ovule. Pollen is particularly sensitive to stress, especially in monocots. In this work, plants were stressed after male and female gametophyte development was complete, and the effect on subsequent reproduction was evaluated. Pollination and fertilization rates were examined to determine the effect that the male gametophyte had on fertility.

Previous work showed that the expression of a group of ROS-scavenging genes was significantly altered in Arabidopsis ovules following salt stress [28]. Here we analyze ROS-scavenging mutants that were differentially expressed in stressed ovules at a critical stage of ovule development. We hypothesize that these ROS-scavenging genes may regulate the ROS levels during ovule abortion. In this study, we characterized the role of three *PER* genes in regulating ROS levels in Arabidopsis ovules, and we tested whether mutants of these loci affect ROS accumulation in ovules or seed failure.

## 2. Results

### 2.1. Salt Stress Lowers Floral Water Potential

After male and female gametophyte development was complete, plants were stressed to determine the downstream effects on reproduction. Since flowers and gametophytes develop synchronously, the gametophyte stage can be inferred from the floral phenotype [29]. Flowering plants were watered once with 75 mM NaCl, the solution was absorbed until the soil reached 100% soil moisture content, and the excess salt solution was discarded. To determine how this osmotic stress affected flowers, inflorescence water potential was measured. Within 6 h, the water potential in flowers dropped significantly and fell to maximal levels within 3 days (Figure 1). Plants were watered every other day, so the water potential data oscillated as the soil dried and rehydrated (Figure 1). These measurements showed that watering with salt increases the number of solutes in the soil and lowers inflorescence water potential for days. Water potential could drop as a result of increased ion transport and accumulation in flowers or a drop in cell turgor in this region. Previous analyses showed that salt stress caused sodium ions to increase by 25% in flowers, but this increase was offset by a decrease in potassium ions [26], resulting in little change in flower osmotic potential. In flowers, salt stress primarily affected water potential and cell turgor.

### 2.2. Increased Ovule Abortion Rate in Peroxidase Mutants

Fitness was evaluated in five ROS-scavenging genes (*per17, 28, 29, apx4*, and *fsd2*). RT-PCR showed that no full-length transcripts were present in the RNA population of leaves and flowers (Figure 2). Previous work identified these genes as differential during ovule abortion [28]. Since the mutation lesion in each of these mutant lines occurs within or upstream of the peroxidase or SOD domain, these are all loss-of-function mutants (Figure 2). Previous research showed that stage 12 flowers are especially susceptible to environmental stress [26]. Therefore, stage 12 flowers were marked, and the fertility was measured after plants were either salt-stressed treated with 75 mM NaCl for 48 h or under healthy growth conditions. Under normal growth conditions, three class III peroxidase mutants (*per17*, *per28*, *per29*) and a superoxide dismutase mutant (*fsd2*) caused significant increases in the ovule abortion rates (Table 1). The *apx4* mutant did not significantly affect fertility. Except for *fsd2,* the stress treatment significantly altered the fertility of all genotypes. There was a highly significant interaction between the treatment and genotype when comparing the fertility of *fsd2* with wild-type plants so, according to Seltman [30], the *p* values for the main effects (genotype and treatment) are ignored and assumed to be statistically significant. The double and triple mutants of *PER* genes were created, and the ovule abortion rates of these mutants also rose significantly. Ovule abortion rates were also scored for Arabidopsis under 75 mM NaCl salt stress for 48 h. The ovule abortion rates increased significantly in *per17* mutant, *per17per28* and *per17per29* double mutants, and *per17per28per29* triple mutants (Table 1). 

In order to compare the fertility and abortion rates in different experimental replicates, the fertility of each mutant was normalized with wild-type controls that were grown in the same trays as the experimental plants. For the wild-type controls, 95% of the ovules successfully set seed under healthy growth conditions; for the salt-stressed controls, 55% of the ovules set seed. When compared to reproduction rates in the respective single mutants, relative fertility decreased even more in salt-stressed *per17*, *per17per28* and *per17per29* double mutants, and the *per17per28per29* triple mutant (Table 1). Notably, the ovule abortion rates of these mutants significantly varied from those of wild-type fruits following salt stress (Table 1), indicating that their mutants responded more sensitively to salt stress. These results concur with the differential expression data reported by Sun et al. [28]: mutation of genes that were induced by environmental stress led to larger fertility effects when the plants were exposed to salt stress. Conversely, the fitness of mutants in genes that were most abundant in unstressed plants showed greater effects in healthy plants.

### 2.3. Fertilization Rates and Seed Failure

Images of aborting ovules are shown (Figure 3). After salt stress, Sun et al. [28] reported that ROS were first detected in the gametophyte. As ovule abortion progressed, ROS were found throughout the ovule [28]. To determine whether or not salt stress correlates with the formation of ROS in ovules, samples from stressed plants were stained with CH_2_DCFDA, a ROS-sensitive dye. As opposed to traditional histochemical stains, nitroblue tetrazolium or cerium chloride that specifically measure superoxide or peroxide levels, CH_2_DCFDA interacts with a wide variety of ROS molecules with different affinities so fluorescence of this dye yields a qualitative summation of ROS. In each *per* mutant genotype, ROS accumulation was evaluated in ovules 48 h after salt stress (Figure 4). We observed significantly higher levels of ROS accumulation in the ovules of *per17, 28*, and *29* single mutants than in wild-type controls (*p* <0.0001). Even greater ROS levels were detected in the double and triple mutants.

### 2.4. H_2_O_2_ Levels in Pistils

Peroxidases neutralize H_2_O_2_ so the amount of this metabolite was measured in pistils. Pistils were used because this was the smallest segment of the plant that could be dissected from without producing wound-induced ROS. ROS levels increase in ovules (Figure 5), but low levels are detected in the carpel walls, stigma, and style. Measuring the level of H_2_O_2_ in *per* mutants will help determine whether or not the accumulation of ROS in ovules is a result of the production of H_2_O_2_ due to a reduction in peroxidase activity. A coupled enzyme assay using a fluorescent indicator was used to detect H_2_O_2_ levels in pistils. Results show H_2_O_2_ accumulated in the pistils of peroxidase triple mutants was significantly higher than in those of wild-type controls in both unstressed and salt-stressed conditions (Figure 5a). While H_2_O_2_ levels in *per* single and double mutants increased, they were not significantly higher than controls. Plant tissues contain many peroxidase isoforms, so the removal of one is likely to have an incremental effect.

### 2.5. Myc-Tagged Peroxidases Exhibited Peroxidase Activity

Peroxidases that have an N-terminal myc-tag were ectopically expressed in Arabidopsis. Myc-tagged PER17, 28, and 29 were fished from total protein extracts using an anti-myc antibody that coated the ELISA plate wells. The peroxidase activity of the tagged proteins was examined. Results showed that myc-tagged PER17, PER28, and PER29 showed significantly more peroxidase activity than the control that contained no myc-tagged protein (Figure 5). This indicates that myc-tagged proteins contain active peroxidases that neutralize H_2_O_2_.

### 2.6. Expression and Sub-Cellular Localization of Peroxidases

While *PER17*, *PER28*, and *PER29* mRNA were detected in many plant tissues, quantitative PCR results revealed that they were most abundant in pistils (Figure 6). The fact that these three *PER* mRNA levels are low in leaves may explain why there was no apparent mutant phenotype in vegetative tissues. To check whether other class III peroxidases might compensate for the loss of functions in mutant alleles, RNA levels for these genes abundant in stage 12 pistils were evaluated. The expression of each class III peroxidase was evaluated using the eFP browser [29]; *PER9*, *PER39*, and *PER68* transcripts are relatively abundant in stage 12 pistils, the organ evaluated in this study. In the stressed *per17*, *per28*, and *per29* single mutants, quantitative RT-PCR showed that the expression of the other five genes remained steady—none of the genes significantly differed from controls.

Many peroxidases are active in peroxisomes, so the PTS1 predictor [31] was used to evaluate the putative *PER17*, *PER28*, and *PER29* protein sequences. This bioinformatics analysis revealed a low probability that these proteins target peroxisomes. Constructs containing GFP translational fusion protein were created to infer the sub-cellular localization of each peroxidase. Transgenic plants containing peroxidase-GFP proteins were examined by confocal microscopy. Results showed that three peroxidase proteins were present in the cell wall or apoplast (Figure 6).

### 2.7. Promoter Activity of Peroxidases in Reproductive Tissues

To investigate the spatial expression of three PERs, Arabidopsis plants that were transformed with *PER::GUS* constructs were analyzed. Results revealed that the *PER17* promoter was active in sepals, stamen, nectaries, style, and the chalaza region of ovules (Figure 6e,f). In some ovules, GUS staining also appeared near the synergids and filiform apparatus of the *PER17::GUS* plants (Figure 6f). In the *PER28::GUS* plants, this gene was first expressed in the transmitting track of stage 12 flowers (Figure 6g). In stage 13 flowers, when the stigma is most receptive to pollination, *PER28* displayed maximal expression (Figure 6h). Plants that carried the *PER29::GUS* construct showed GUS staining in ovules, especially around the gametophyte (Figure 6i,j). In the siliques, *PER29::GUS* staining was only found in ovules but not developing seeds, indicating that the *PER29* promoter is active in gametophytes but not embryos. In addition, *PER29* was expressed in guard cells (Figure 6k). For each of these constructs, stems, rosette leaves, and cauline leaves were stained, but no signal was observed when floral parts were fully stained. Evaluation of eFP expression data for these constructs [32] indicates that these genes are expressed in other regions of the plant but at much lower levels, so the GUS signal appeared absent when the signal from reproductive tissues was observed.

## 3. Discussion

### 3.1. Peroxidase Mutants and Fertility

The genes in this study were chosen for analysis because they exhibited significant changes in expression at the initiation of ovule abortion. These ROS-scavenging genes were *PER17*, *PER28*, *PER29*, *APX4*, and *FSD2*. While *PER17* expression increased, the levels for the other four genes were significantly lower following salt stress [26]. Previous investigation correlated ROS accumulation in ovules with the rate of ovule abortion [33]. Results show that mature gametophytes or developing ovules tolerate up to 8 h of 75 mM salt stress with minimal disruption to reproduction, but treatment with 200 mM NaCl for 12 h causes more than 90% of the ovules to abort [26]. From a global microarray study [28], five ROS scavengers that exhibited significant changes in gene expression at the critical developmental stage were selected for further analysis.

Wide-scale mutational analysis revealed that organisms contain a small fraction of essential genes, but most genes were genetically redundant or exhibited distributed robustness [34,35]. Recent analysis indicates that the vast majority of genes serve a function, but these include chromatin effects, RNA interference, DNA modification, and components altering the genetic landscape (Encode Consortium, 2012). Partial redundancy and distributed robustness both lead to incremental phenotypes instead of lethality. While *APX4* transcripts are abundant in leaves and developing embryos [32], mutation of the *APX4* locus had no effect on ovule development or plant fertility (Table 1). The absence of an ovule or seed phenotype in *apx4* mutants may be due to genetic redundancy with another ROS scavenger. ROS scavengers that are expressed in developing seeds can limit oxidative stress during seed desiccation and germination, thereby reducing seed deterioration [36]. Mutation of the *APX4* locus primarily affects seedling growth and establishment [37].

### 3.2. Stress, Photosynthesis, and ROS Scavengers

The rate at which stress is applied to plants affects the physiological outcome. When water stress slowly increases, plants activate different regulatory networks, permitting acclimation and tolerance to water stress [38]. Once plants in this study were stressed, the rapid change in floral water potential allowed little time for this type of acclimation. Weeks after the imposition of stress, we observed a higher fraction of ovules formed seeds. This observation indicates plants acclimated to these soil conditions.

Myouga et al. [39] reported that FSD2 is active in chloroplasts. When plants were grown at five-fold higher fluence rates than those described here, *fsd2* leaves were chlorotic. At lower fluence rates, the other two Arabidopsis SOD loci were sufficient to protect chloroplasts from photo-oxidation. Reproductive analyses of *fsd2* mutants revealed a significant reduction in fertility for actively photosynthesizing plants but had minimal effect on stressed plants (Table 1). The rather small effect of this mutation on fertility after stress demonstrates the reduction in photosynthetic rates creates less superoxide production in the chloroplasts. The absence of a fertility phenotype indicates that the other two SODs work redundantly and scavenge the residual superoxide generated.

### 3.3. Peroxidases Affecting Fertility

In this study, we measured how three class III peroxidase loci affected ROS metabolism and ovule abortion. The nomenclature of class III peroxidases in *Arabidopsis* varies, where both PER or PRX have been used. In earlier studies, PER17 was reported to modulate lignification and pod shatter [19], and PER28 was hypothesized to affect pollen-pistil interactions [40]. We found that PER17, PER28, and PER29 affected H_2_O_2_ production in ovules. Mutation of individual loci showed modest increases in peroxide accumulation (Figure 5). The significant increase in ROS shown in Figure 4 indicates these pistil-specific peroxidases work together to cause this increase in activity.

The level of *PER17* mRNA increased after 24 h of stress, while the expression of *PER28* and *PER29* was repressed under these conditions (Figure 3C). Following salt stress, *per17*, *per17per28*, *per17per28*, and *per17per28per29* mutants exhibited significantly lower fertility (Figure 3B) than unstressed counterparts (Figure 3A). One simple explanation for this is that increased *PER17* activity after salt stress limits ovule failure. Conversely, *per28, per29*, and *per28per29* mutants showed significant ovule abortion rates in healthy plants, but not after environmental stress (Table 1), indicating that *PER28* and *PER29* scavenge peroxides in healthy plants but have negligible activity in those that are salt-stressed. These data suggest that under normal conditions, all three *PER* genes scavenge ROS in ovules or nearby areas to which this molecule can diffuse, thereby protecting ovules from abortion. When encountering salt stress, *PER17* remains an influential peroxide scavenger that removes ROS.

We observed ROS accumulation after salt stress in ovules of three peroxidase mutants (Figure 4), which indicated that either the rate of ROS removal decreased or its synthesis accelerated. ROS accumulation further increased in *per* double and triple mutants (Figure 4). However, the ovule abortion rates in some of the peroxidase mutants were not affected. We cannot distinguish which types of ROS accumulated in these tissues because CH_2_DCFDA stains many ROS. One or more types of ROS may reach a critical threshold, signaling the described physiological changes. Data reported here show that the induction of ROS accumulation by the mutation of peroxidases was sufficient to increase the rate of seed abortion. 

It is believed that H_2_O_2_ may be a better signaling molecule than other types of ROS because it can cross the plasma membrane and move into neighboring cells. In plants, class III peroxidases can reduce peroxides by using various donor molecules, such as auxin or secondary metabolites (e.g., lignin precursors and phenolic compounds) [19]. Some cell wall peroxidases modulate ROS levels, which affect plant defense or development, but this reaction is strongly pH-dependent [24,41,42,43]. Therefore, we quantified H_2_O_2_ levels in our peroxidase mutants. The H_2_O_2_ levels in peroxidase single and double mutants were higher than the wild-type controls but were not significantly higher as a result of the high variance. For given genotypes, this variance can be explained by variability in the activity in the family of 70+ peroxidases present in Arabidopsis, which might or might not be due to differences in ovule abortion rates among pistils. Purification of H_2_O_2_ is labor intensive, and samples cannot be stored frozen, so the sample size for each genotype was limited. The hydrogen peroxide levels were significantly higher in triple mutants under both unstressed (*p* < 0.05) and salt-stressed (*p* < 0.01) conditions (Figure 5). These data correlated with low fertility (Table 1 and Figure 3), indicating H_2_O_2_ may be the molecule that causes ovule abortion.

According to the expression patterns from GUS staining results, three Class III peroxidases may modulate ROS signaling and/or affect the fertilization process, nutrient transport, and/or pistil growth. *PER29::GUS* revealed that this gene is active in ovules, especially around the gametophyte (Figure 6j). Sun et al. [28] reported that after salt stress, ROS initially accumulated in the gametophyte and then spread throughout the ovule. Thus, *PER29* may be responsible for regulating ROS in the gametophyte prior to fertilization. Together with the fact that *PER29* has high levels of expression in healthy controls (Figure 3C), these results explain why the *per29* mutants exhibited the lowest relative fertility among the genotypes tested (Figure 3A).

The tip growth of pollen tubes requires ROS buildup and the presence of an antioxidant that alters pollen growth in vitro [44]. Before fertilization, a pollen grain adheres to the stigma, and the pollen tube emerges and grows along the transmitting tract, which contains nutrients and signaling molecules that affect pollen tube guidance. Therefore, the fertilization of *per17*, *28,* and *29* mutants was examined. Flowers were emasculated, and pollen tubes were stained 24 h after manual pollination [45]. After 24 h, pollen tubes entered 90% of the wild-type, *per28*, and *per29* ovules, but only 50% of the *per17* pollen tubes reached an ovule. Since *per17* had 70% fertility (Table 1), we conclude that pollen tube growth occurs more slowly or pollen tube guidance is disrupted in this genotype. Wild-type pollen fertilized significantly fewer *per17* ovules, which is consistent with a pollen guidance problem. Future experiments are necessary to discriminate between these possibilities.

*PER28::GUS* was expressed in the transmitting track (Figure 6g). When the stigma is most receptive to pollination, *PER28* displayed maximal expression in the stigma (Figure 6H). The expression of these genes is consistent with a role in pollen-stigma interaction or pollen tube growth/tracking. However, the pollination and pollen tube growth appeared normal in the *per28* mutant. During manual pollination of the pistil, it was noted that the pollen did not adhere as well to the stigma, which suggests the *PER28* locus may affect pollen adhesion to the stigma.

### 3.4. ROS, Fertility, and PCD

In a variety of plant cell types and plant species, PCD is induced with a pulse of ROS. Images in this paper show necrosis of the endothelium of stressed ovules, which coincides with ROS accumulation (Figure 3, Figure 4, Figure 5 and Figure 6). Seed abortion is preceded by changes in mitochondrial potential [33]. Previous work shows that similar peroxidases can regulate apoplastic oxidative burst and then PCD in response to fungal infection [46]. When pollen nuclei are delivered to the synergids by the pollen tube, ROS induce PCD in these two cells [47]. This is mediated by the activity of a kinase MAP kinase cascade, which effects synergid PCD and immunity-associated PCD [48]. In tomatoes, disruption of fertilization leads to altered endothelium development and ectopic ROS spikes. This can produce parthenogenic seeds or altered developmental fates [49,50]. In wheat, drought stress increases ROS production and increases yield over a shortened fertility window [51]. In a number of different plants, ROS affect reproductive development by modulating PCD and changing cell fate.

## 4. Materials and Methods

### 4.1. Plant Growth and Fertility Measurements

*Arabidopsis thaliana* wild-type (Col-0) and ROS scavenger T-DNA insertion mutants (SALK_003180 for *per17*, SALK_076194 for *per28*, SALK_133065 for *per29*, SAIL_519_E04 for *apx4*, and SALK_080457 for *fsd2*) were identified from ABRC stocks (Columbus, OH). After backcrossing these mutant alleles with wild-type Arabidopsis (Col-0), homozygous mutants were isolated by a polymerase chain reaction (PCR). Two gene-specific primers from upstream and downstream coding sequences were used to verify the presence of wild-type alleles. One gene-specific primer and one primer from the T-DNA border sequence were used to confirm the presence of the mutant allele (Appendix A). All mutants are null mutants since none encode a full-length transcript (Figure 2).

Plants were grown in 2 × 2 pots in a Percival plant growth chamber (Perry, IA) at 24 °C, 50% relative humidity, and continuous fluorescent light (100 μmol photon m^−2^ s^−1^). To stress plants, the pots were soaked in irrigated water containing 75 mM NaCl for 6 h, drained, then not irrigated with plain water for an additional 42 h.

Healthy seeds and aborted ovules in each silique (developed from pistil) were recorded. From these data, ovule abortion rates and fertility with standard errors were calculated. Before performing ANOVA analysis using JMP8 software (Cary, NC, USA), ovule abortion rates were root arcsine transformed to generate a normal distribution. A normality test was done in JMP8 for all other ANOVA analyses.

### 4.2. RNA Extraction and PCR

Total RNA was isolated from 10-day-old seedlings and tissues from 30-day-old hydroponic plants using RNeasy mini kit (Qiagen, Valencia, CA, USA). Hydroponic propagation was described by Gibeaut et al. [52]. Complementary DNA (cDNA) was synthesized using SuperScript III reverse transcriptase (Invitrogen, Grand Island, NY, USA) from 1 μg of total RNA as suggested by the manufacturer. Quantitative PCR (qPCR) was performed in optical 96-well plates using an ABI StepOnePlus machine (Applied Biosystems, Grand Island, NY, USA). Each 10-μL reaction consisted of GoTaq qPCR Master Mix (Promega, Madison, WI, USA), 2 μL of 1:50 diluted cDNA, and 0.2 mM each of gene-specific primer pairs (see Appendix A). The following thermal profile was used for qPCR: 95 °C for 2 min; 30 cycles of 95 °C for 15 s and 60 °C for 1 min; and a melt curve analysis at 95 °C for 15 s, 60 °C for 15 s, and 95 °C for 15 s. C_T_ and standard curve were extracted using ABI StepOne v2.2 software (Grand Island, NY, USA). The fold change of gene expression and standard deviations were calculated according to the guide from ABI (http://www.appliedbiosystems.com/absite/us/en/home/support/tutorials.html, accessed on 6 April 2023). Melt curve analyses were used to verify that a single product was produced from each qPCR.

### 4.3. Metabolite Assays

Pistils were dissected lengthwise and incubated in 10 μM 5-(and 6-) carboxy-2′, 7′-dichlorodihydrofluorescein diacetate (CH_2_DCFDA, Molecular Probes, Grand Island, NY, USA) for 30 min, then samples were washed twice in 10 mM of phosphate buffer pH 7.0. In the presence of ROS, CH_2_DCFDA was oxidized to form carboxy-dichloro-fluorescein (CDCF), which is highly fluorescent. The CDCF was visualized by excitation between 490 and 510 nm, and the fluorescent emissions were detected using a 520 nm long-pass filter. This filter set detects both CDCF emission (green) and plant cell auto-fluorescence (red). 

Flowers were frozen in N_2_(l). Using a steel block immersed in N_2_(l) and frozen forceps, pistils were dissected from flowers, and pistil length was measured with a dissecting microscope. After the pistils were ground to powder, 0.1 mL of 0.2 M HClO_4_ was added to stop all protein activity. To ensure that wound-induced H_2_O_2_ was not produced, samples were kept frozen until they were dissolved in HClO_4_. Samples were kept in this buffer at 0 °C for 5 min and then centrifuged at 20,800× *g* for 10 min at 4 °C. The supernatant was neutralized with 110 µL of 0.2 M NH_4_OH and was centrifuged again at 3000× *g*. The supernatant was passed through a column of AG resin (Bio-Rad, Hercules, CA, USA) and then eluted with 0.1 mL deionized water.

Amplex Red Peroxidase Assay Kit (Invitrogen, A22188) was used to quantify H_2_O_2_ levels in the extracts, as recommended by the manufacturer. Fluorescence was measured using Synergy HT fluorescence plate reader (Bio-Tek, Highland Park, IL, USA) with excitation/emission at 540/590 nm. Fluorescence intensity was measured every 10 min to ensure that fluorescence detection of the standards and unknown samples exhibited a linear relationship between accumulation and time. The concentration of H_2_O_2_ in the samples was calculated using a standard curve with HRP of known activity, which was included with the kit as a standard.

### 4.4. Constructs and Plant Transformants

The pEW201ML plasmid was created to contain the PER17 coding sequence (upstream) that is translationally fused to the GFP coding sequence (downstream) and is driven by the CAMV 35S promoter. Similarly, the pEW202ML and pEW203ML binary plasmids were created to contain PER28 and PER29 with translationally fused GFP, respectively. Plants were transformed with this *A. tumefaciens* containing the plasmid, according to Clough and Bent [53]. Transformants were selected by spraying 1000-fold diluted BASTA (Finale, AgrEvo, Pikeville, NC, USA).

To synthesize proteins for the assay, PER17, 28, and 29 genes were ectopically expressed using CaMV 35S promoter. The pEW401ML plasmid (*35S::myc-PER17*) contains the PER17 coding sequence and an upstream in-frame myc tag (MEQKLISEEDL). The pEW402ML and pEW403ML plasmids contain, respectively, PER28 and PER29, each with a myc tag. Transformants containing each construct were generated as described above. Myc-tagged protein expression was confirmed in these transformants (Appendix A). Total soluble proteins were extracted from 12-day-old seedlings. The extraction buffer contained 50 mM Tris-HCl pH 8.5, 1X protease inhibitor cocktail VI (RPI Corp), and 0.2 M β-mercaptoethanol. Prior to binding, each well of the FLUOTRAC 600 immunology plate (Greiner Bio-One, Monroe, NC, USA) was coated with a 0.5 μg myc-tag antibody (Millipore, Billerica, MA) at 4 °C overnight as described by Pratt et al. [54]. After washing and blocking, myc-tagged proteins were fished from total protein extracts (1 mg of myc-PER17; 0.9 mg of myc-PER28; and 0.8 mg of myc-PER29). Following additional washes, Amplex Red was applied to determine the activity of myc-peroxidases as recommended by the manufacturer. Fluorescence was measured with excitation/emission at 540/590 nm. The relative fluorescence of each myc-peroxidase was normalized with wild-type controls in each experimental replicate.

To construct *PER17::GUS*, 1.9 kbp of *PER17* upstream sequence was PCR amplified and transcriptionally fused to the upstream of the *uidA* gene start codon [55]. *PER28::GUS* and *PER29::GUS* constructs were created similarly. GUS assays were done as previously described [56].

## 5. Conclusions

Here we report three loss-of-function peroxidase mutants (*per17*, *per28*, and *per29*) were found to accumulate more ROS and H_2_O_2_ in ovules, and the ovule abortion rates significantly increased in these mutants. A simple hypothesis explaining the results reported here is that PER17 and PER29 proteins, which are predominantly expressed in ovules, cause oxidative bursts in ovules, inducing ovule abortion. 

## Figures and Tables

**Figure 1 plants-12-02182-f001:**
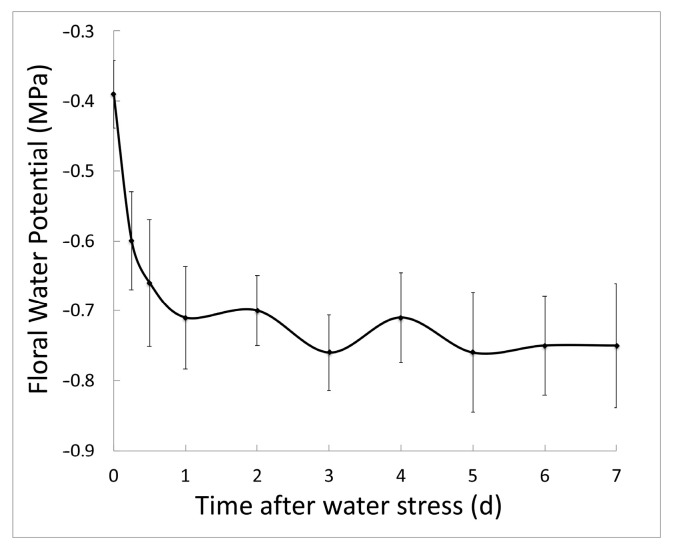
Using a pressure chamber, the water potential of inflorescences was measured periodically after plants were watered once with 75 mM NaCl. The initial point shows the water potential in healthy plants. Following stress, water potential decreases and plateaus after a day. Plants were watered on day 2, 4, and 6, so a slight rise in water potential was observed on those days. Five or more inflorescences were measured at each time point. The best-fit line through the average at each time is plotted. Average ±1 standard deviation shown.

**Figure 2 plants-12-02182-f002:**
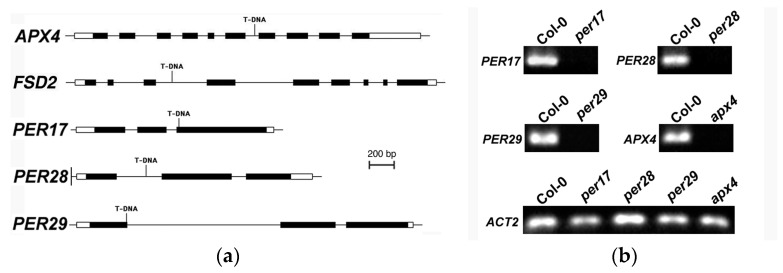
Analysis of ROS scavenger mutants. (**a**) Molecular models of *APX4, FSD2, PER17, PER28, and PER29*, including the T-DNA insertion sites, are shown. Genes are shown 5′ (left) to 3′ (right). Solid boxes denote exons, lines represent introns, and open boxes indicate untranslated regions. (**b**) RT-PCR analysis revealed that *per17*, *per28*, *per29*, and *apx4* homozygous mutant alleles contain no detectable full-length transcripts. *per17*, *per28*, *per29*, and *apx4* are null alleles. *ACTIN2* (*ACT2*) served as a positive control.

**Figure 3 plants-12-02182-f003:**
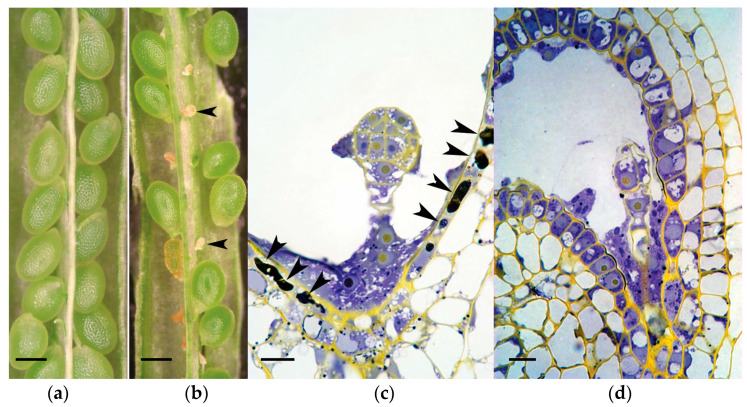
Representative images of dissected pistils of (**a**) healthy and (**b**) salt-stressed plants. Two aborting/aborted ovules were indicated (arrowheads). Globular embryos from (**c**) stressed and (**d**) healthy ovules are shown. In the stressed ovule, most of the endothelium cells were necrotic and had degenerated (arrowheads). In the healthy control, endothelial cells were cytoplasmically dense, indicative of high metabolic activity associated with the movement of nutrients and metabolites from the maternal plant into the embryo sac. Size bars are 100 µm (**a**,**b**) and 10 µm (**c**,**d**).

**Figure 4 plants-12-02182-f004:**
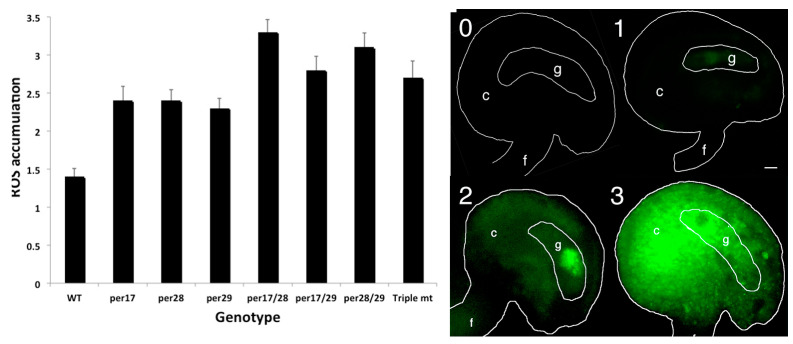
Following salt stress, peroxidase mutants accumulated ROS in ovules. Prior to stress, stage 12 flowers were marked. CH_2_DCFDA fluorescence intensity in ovules was evaluated: level 0 had no detectable ROS; level 1 had detectable ROS in the embryo sac or gametophyte (g); level 2 had ROS accumulation in the chalaza (c); and level 3 had copious ROS accumulation throughout the ovule. All *per* mutants had significantly more ROS than controls (*p* < 0.01). ANOVA analyses were done using JMP8 (unequal variances and two-tailed distributions). Size bar is 10 µm.

**Figure 5 plants-12-02182-f005:**
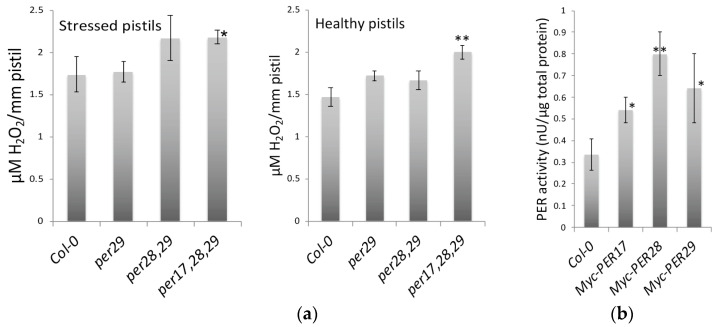
Peroxidase activity and H_2_O_2_ accumulation in *per* mutants. (**a**) The amount of hydrogen peroxide was compared between wild-type pistils and peroxidase mutant pistils. Healthy and stressed stage 12 Arabidopsis flowers were marked. Plants were treated with either 75 mM NaCl (stressed) or water (healthy). After 48 h, H_2_O_2_ was measured in five or more pistils from each genotype and treatment. Asterisks indicated significant differences between the mutant genotypes and the wild-type (Col-0) controls (* *p* < 0.05 and ** *p* < 0.01) (**b**) Peroxidase activity from myc-PER17, myc-PER28, and myc-PER29 transformants was measured. Each well of an ELISA plate was coated with a myc-tag antibody so that myc-tagged proteins could be fished from total protein extracts isolated from plants. Amplex Red substrate was used to determine the activity of these peroxidases. Peroxidase activity from these transformants was significantly greater than the wild-type controls (* *p* < 0.05 and ** *p* < 0.01). ANOVA analysis was done using JMP8 (unequal variances and two-tailed distributions).

**Figure 6 plants-12-02182-f006:**
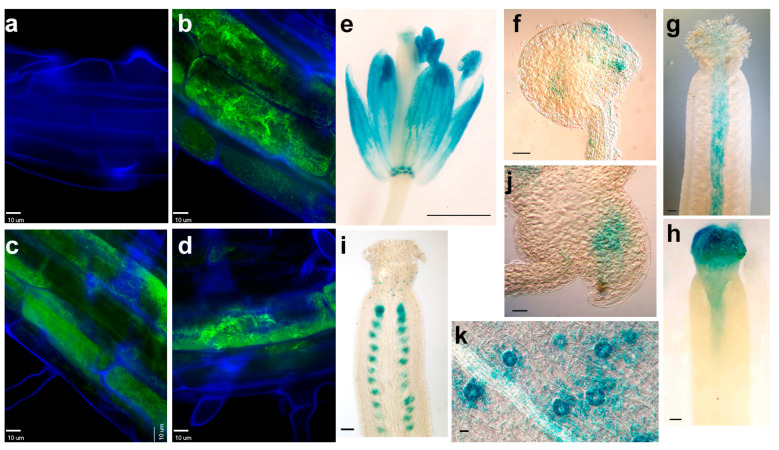
The expression of three PER proteins. Merged confocal images show GFP (green) and cell wall (blue) fluorescence in (**a**) wild type, (**b**) PER17-GFP, (**c**) PER28-GFP, and (**d**) PER29-GFP cells. GUS histochemical staining of *PER17::GUS* (**e**,**f**), *PER28::GUS* (**g**,**h**), and *PER29::GUS* (**i**–**k**) plants is shown. PER17 promoter was active in flowers (**e**) and the chalaza region of ovules (**f**) and leaves. *PER28* was initially expressed in the transmitting tract of stage 12 pistils (**g**) and increased in abundance in the stigma of stage 13 pistils (**h**). *PER29* promoter was active in ovules (**i**) and sepal guard cells (**k**). A higher magnification of panel i shows staining in the gametophyte (**j**). (**a**–**d**,**f**,**j**,**k**), bar = 10 μm; (**e**), bar = 1 mm; (**g**–**i**), bar =100 μm.

**Table 1 plants-12-02182-t001:** Mutation of ROS-scavenging genes significantly reduced fertility. Prior to stress, stage 12 flowers were marked, and the subsequent ovule abortion rates for these fruits determined. Stressed plants were treated with 75 mM NaCl for 48 h. For each genotype, 30 pistils were scored. After arcsine transformation, ANOVA tests compared fertility between mutant genotypes and grouped controls: differences of less than 0.05 ^a^, 0.01 ^b^, 0.001 ^c^, and 0.0001 ^e^ indicated.

	Abortion Rate (%)	
Genotype	Healthy ^1^	Stressed ^1^
Wild type	12.9 ± 2.0	26.9 ± 4.6
*per17*	29.6 ± 5.8 ^b^	44.6 ± 6.0 ^a^
*per28*	23.2 ± 3.3 ^b^	34.6 ± 4.4
*fsd2*	35.2 ± 6.8 ^c^	29.0 ± 4.2
*apx4*	21.4 ± 4.2	23.2 ± 4.0
Wild type	18.0 ± 1.7	33.4 ± 3.3
*per29*	50.8 ± 3.5 ^e^	41.6 ± 4.8
Wild type	14.9 ± 1.3	33.1 ± 2.7
*per17 per28*	39.7 ± 6.3 ^c^	64.0 ± 6.6 ^e^
Wild type	23.9 ± 3.1	49.5 ± 4.5
*per17 per29*	35.6 ± 6.0	67.7 ± 6.9 ^b^
Wild type	21.6 ± 1.5	39.5 ± 3.6
*per28 per29*	33.7 ± 4.0 ^b^	40.0 ± 2.9
Wild type	5.4 ± 0.7	25.4 ± 4.6
*per17 per28 per29*	31.5 ± 5.2 ^e^	66.3 ± 5.6 ^e^

^1^ The ovule abortion rate is the average ± one standard error.

## Data Availability

Mutants are available from the Arabidopsis Biological Resource Center. Constructs are available from the corresponding authors by request.

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
