# Peer review of "During Water Stress, Fertility Modulated by ROS Scavengers Abundant in Arabidopsis Pistils"

_plants, 2023, doi:10.3390/plants12112182_

Round 1
Reviewer 1 Report
Dear authors,
The general scope and intended objectives of the paper are clear and in line with the given topic. Some insightful information especially on ROS-scavenging genes and how they regulate the ROS levels during ovule abortion in healthy and stressful conditions has been adequately presented.
The study was correctly designed and technically sound. The methods and reagents were described with sufficient details to allow another researcher to reproduce the results. The results were interpreted appropriately and significant. The conclusions were justified and supported by the results. In addition, the English language was appropriate and understandable.
This study merits to be published after revision and fixing these two issues found in your paper:
- Paragraph section 2.3 in results: missing information about H2O2 level. Redundant with section 2.4 from results. Please fix this issue.
- Figure 5-b: missing asterisks for significance level. Please fix this issue.
Minor editing are required.
Author Response
There were two issues that you raised in your review.
The first was a problem copy/pasting the word file into the Plants template. I deleted a section of text and duplicated a section of the text when I formatted the submission. The duplicated portion was removed and the original text was inserted.
I added significance symbols to Fig. 5b. I modified the figure legend to account for the P<0.01 symbol in this panel.
Thank you for finding both of these problems and bringing them to my attention!
Reviewer 2 Report
The manuscript During water stress, fertility modulated by ROS scavengers abundant in Arabidopsis pistils Ya-Ying Wang, DJ Head, Logan Peoples, Bernard A Hauser discusses the involvement of peroxidases and ferum-dependent superoxide dismutase in the regulation of fertility. The topic of this experimental article is in the active interest of many researchers. The paper also discusses the role of progamated cell death with an increase in ROS.
This manuscript contains all the necessary parts and is designed according to the rules of the MDPI, with the exception of minor design features.
Text:
In the introduction, the last paragraph, instead of the formulated goal and objectives, contains a statement. In the literary genre of a scientific article, intrigue should remain, since this is not a thesis!
In conclusion, the authors discuss the results and even posted a link, while there should be clear and concise conclusions and prospects for their good research.
Methodologically, it is not clear to me how the authors confirm the PCG.
Figure 5 with non-arrowed necrosis and the statement "endothelium cells were necrotic and the endosperm"
nuclei appeared to have degenerated" given that they are not marked and not distinguishable, they look unconvincing, I do not see nuclei - enlarge the image, indicate what you describe and reformulate more accurately.
By introduction and discussion. This topic can be said to be in the frontier, but not in the aspect of Arabidopsis. I think we should add a discussion of PCG in the aspect of AFC with examples of modern articles on cultivated plants, this will expand the audience of the article from theoretical and fundamental to practical applications. For discussion: Hauser B. A. et al. Changes in mitochondrial membrane potential and accumulations of reactive oxygen species precede ultrastructural changes during ovule abortion //Planta. - 2006. - T. 223. - S. 492-499 .; Khaliluev M. R. et al. Abnormal floral meristem development in transgenic tomato plants do not depend on the expression of genes encoding defense-related PR-proteins and antimicrobial peptides // Russian journal of developmental biology. - 2014. - T. 45. - S. 22-33 .; Volz R. et al. ROS homeostasis mediated by MPK4 and SUMM2 determines synergid cell death //Nature Communications. - 2022. - T. 13. - No. 1. - S. 1746.; Baranova E. N. et al. Possible role of crystal-bearing cells in tomato fertility and formation of seedless fruits // International Journal of Molecular Sciences. - 2020. - T. 21. - No. 24. - S. 9480.; Hu C. H. et al. Interaction between TaNOX7 and TaCDPK13 contributes to plant fertility and drought tolerance by regulating ROS production // Journal of Agricultural and Food Chemistry. - 2020. - T. 68. - No. 28. - S. 7333-7347.; Liu Y. et al. Reactive oxygen species accumulation strongly allied with genetic male sterility convertible to cytoplasmic male sterility in kenaf //International Journal of Molecular Sciences. - 2021. - T. 22. - No. 3. - S. 1107.
It is somewhat surprising that the authors do not confirm PCG by other available methods than light microscopy, this is quite simple and will improve the article. If this is difficult, enlarge the image and indicate what proves the statement.
Pictures:
So figures 1, 4 contain fragments of a frame that does not correspond to the design style.
In Figure 2, the regions of introns and exons are not labeled, which makes it impossible to evaluate both the significance and/or make inquiries in the appendix or literature.
Figures 3 and 4 have strange ways of captioning images: in one case, these are letters in brackets and with a dot, in the second, numbers. I think it needs to be consistent. Scale bars are missing in both figures. Letters with designations are very small and almost indistinguishable, it is not clear why there are no the same dark field shots or similar ones where the structures would be visible.
Figure 5 again requires uniformity, the confidence bars in figures (!?) 5a are not visible, the captions under the columns and the sizes of the histograms must also be in the same style. I note that the general title of figure 5 is generally absent.
Figure 6 uniformity of rulers with magnifications is required, why are they of different thicknesses?
I consider the work well done and interesting, and I think that it can be accepted after this small edit and answers to questions.
Author Response
In the introduction, the last paragraph, instead of the formulated goal and objectives, contains a statement. In the literary genre of a scientific article, intrigue should remain, since this is not a thesis!
This paragraph was modified to describe the hypothesis for this paper and question addressed.
In conclusion, the authors discuss the results and even posted a link, while there should be clear and concise conclusions and prospects for their good research.
The citation was removed and the conclusions shorted to a single statement.
Methodologically, it is not clear to me how the authors confirm the PCG.
Figure 5 with non-arrowed necrosis and the statement "endothelium cells were necrotic and the endosperm"
The necrotic endotheilial cells were identified with arrows.
nuclei appeared to have degenerated" given that they are not marked and not distinguishable, they look unconvincing, I do not see nuclei - enlarge the image, indicate what you describe and reformulate more accurately.
This statement was removed and a citation with these observations added to the text.
By introduction and discussion. This topic can be said to be in the frontier, but not in the aspect of Arabidopsis. I think we should add a discussion of PCG in the aspect of AFC with examples of modern articles on cultivated plants, this will expand the audience of the article from theoretical and fundamental to practical applications. For discussion: Hauser B. A. et al. Changes in mitochondrial membrane potential and accumulations of reactive oxygen species precede ultrastructural changes during ovule abortion //Planta. - 2006. - T. 223. - S. 492-499 .; Khaliluev M. R. et al. Abnormal floral meristem development in transgenic tomato plants do not depend on the expression of genes encoding defense-related PR-proteins and antimicrobial peptides // Russian journal of developmental biology. - 2014. - T. 45. - S. 22-33 .; Volz R. et al. ROS homeostasis mediated by MPK4 and SUMM2 determines synergid cell death //Nature Communications. - 2022. - T. 13. - No. 1. - S. 1746.; Baranova E. N. et al. Possible role of crystal-bearing cells in tomato fertility and formation of seedless fruits // International Journal of Molecular Sciences. - 2020. - T. 21. - No. 24. - S. 9480.; Hu C. H. et al. Interaction between TaNOX7 and TaCDPK13 contributes to plant fertility and drought tolerance by regulating ROS production // Journal of Agricultural and Food Chemistry. - 2020. - T. 68. - No. 28. - S. 7333-7347.; Liu Y. et al. Reactive oxygen species accumulation strongly allied with genetic male sterility convertible to cytoplasmic male sterility in kenaf //International Journal of Molecular Sciences. - 2021. - T. 22. - No. 3. - S. 1107.
It is somewhat surprising that the authors do not confirm PCG by other available methods than light microscopy, this is quite simple and will improve the article. If this is difficult, enlarge the image and indicate what proves the statement.
These citations were added to the text. I was unable to address the PCG effects. The only meaning I know for this abbreviation is Polycomb Group protein. I asked the editor for help, but he was unable to help with PCG or AFC abbreviations. I did add in these references to show that ROS effects fertility in other angiosperms. I believe this is what the reviewer is suggesting.
Pictures:
So figures 1, 4 contain fragments of a frame that does not correspond to the design style.
These figures were updated and the frames around the graphs eliminated.
In Figure 2, the regions of introns and exons are not labeled, which makes it impossible to evaluate both the significance and/or make inquiries in the appendix or literature.
I added directionality to the figure legend so reader knows where the genes start and end. Since the exons and non-coding regions are displayed graphically, numbers labeling each exon and intron are not needed and would clutter this figure.
Figures 3 and 4 have strange ways of captioning images: in one case, these are letters in brackets and with a dot, in the second, numbers. I think it needs to be consistent. Scale bars are missing in both figures. Letters with designations are very small and almost indistinguishable, it is not clear why there are no the same dark field shots or similar ones where the structures would be visible.
Scale bars were added to Figures 3 and 4. The journal labels panels with letters in parentheses (no dot).
Figure 5 again requires uniformity, the confidence bars in figures (!?) 5a are not visible, the captions under the columns and the sizes of the histograms must also be in the same style. I note that the general title of figure 5 is generally absent.
The title for this figure was added and panel (a) re-formatted to conform to panel (b).
Figure 6 uniformity of rulers with magnifications is required, why are they of different thicknesses?
I consider the work well done and interesting, and I think that it can be accepted after this small edit and answers to questions.
This figure was modified to make the size bars uniform in size. The lettering was changed to lowercase to be consistent with the journal.
Reviewer 3 Report
Salt stress can lead to seed failure. In this study, authors found that the accumulation of ROS in ovules is associated with seed failure. Further, they found that the ROS scavengers showed differentially expressed in stressed ovules regulate the accumulation ROS and associate with seed failure.
My comments:
1 In Fig 1, statistical analysis should be added.
2 In Figure 4 and Figure 5, please show the methods used for statistical analysis.
3 In “2.3. H2O2 Levels in Pistils”. Authors analyzed the peroxidase activity of the myc-tagged PER17, PER28 and PER29 in vitro. They did not detect the H2O2 levels in pistils. Thus, the title of 2.3 don’t match up the results.
Author Response
This reviewer requested that additional information about the statistical analysis get added to Figures and a section of text that did not match it the heading. The corrections this reviewer requested (ital) and our response are listed below.
In Fig 1, statistical analysis should be added.
- Figure 1 shows a best fit through the points on a water potential curve after stress induction. This function can be selected in excel plots to fit a curvy line through all points in a plot. After stress induction, the water potential oscillates every two days because plants were watered on this schedule. This observation was noted in the legend, as well as additional information on how these data were gathered.
In Figure 4 and Figure 5, please show the methods used for statistical analysis.
- Additional information was requested for the ANOVA tests done for Figures 4 and 5. The distributions of all samples were tested to check if they were normal. Since they all were, we proceeded with the ANOVA analyses using JMP8. A sentence was added to the methods indicating that we did the normality tests for all these data. All of the distributions were found to be two-tailed. For both figures, additional details on the ANOVA analyses were added.
In “2.4. H2O2 Levels in Pistils”. Authors analyzed the peroxidase activity of the myc-tagged PER17, PER28 and PER29 in vitro. They did not detect the H2O2 levels in pistils. Thus, the title of 2.4 don’t match up the results.
- When formatting the submission into the Plants template, I (BH) made a mistake copying text. The text for section 2.5 was copied into both section 2.5 and section 2.4. I deleted the duplicated text and included the text that was originally meant to go in section 2.4.
Thank you for helping to improve this ms.
Round 2
Reviewer 1 Report
The authors, thankfully, have answered and adjusted the issues mentioned in their paper.
Reviewer 2 Report
Manuscript During water stress, fertility modulated by ROS scavengers abundant in Arabidopsis pistils by authors
Ya-Ying Wang, DJ Head, Logan Peoples, Bernard A Hauser has been corrected. Necessary and possible changes have been made. The manuscript may be published.
I apologize for the mistranslation of the abbreviation, which led to misunderstanding.
Reviewer 3 Report
I have no further comments.